# 110 μm thin endo-microscope for deep-brain in vivo observations of neuronal connectivity, activity and blood flow dynamics

Miroslav Stibůrek[1,5], Petra Ondráčková[1,5], Tereza Tučková[1], Sergey Turtaev[2], Martin Šiler [1], Tomáš Pikálek [1], Petr Jákl [1], André Gomes[2], Jana Krejčí [3], Petra Kolbábková[1], Hana Uhlířová [1] ✉ & Tomáš Čižmár [1,2,4] ✉

Light-based in-vivo brain imaging relies on light transport over large distances of highly scattering tissues. Scattering gradually reduces imaging contrast and resolution, making it difficult to reach structures at greater depths even with the use of multiphoton techniques. To reach deeper, minimally invasive endo-microscopy techniques have been established. These most commonly exploit graded-index rod lenses and enable a variety of modalities in head-fixed and freely moving animals. A recently proposed alternative is the use of holographic control of light transport through multimode optical fibres promising much less traumatic application and superior imaging performance. We present a 110 μm thin laser-scanning endo-microscope based on this prospect, enabling in-vivo volumetric imaging throughout the whole depth of the mouse brain. The instrument is equipped with multi-wavelength detection and three-dimensional random access options, and it performs at lateral resolution below 1 μm. We showcase various modes of its application through the observations of fluorescently labelled neurones, their processes and blood vessels. Finally, we demonstrate how to exploit the instrument to monitor calcium signalling of neurones and to measure blood flow velocity in individual vessels at high speeds.

The evolution of neurophotonics, particularly optogenetics and the methods of deep-brain in vivo imaging, continues to extend the experimental possibilities with which one may surveil and control the behaviour of neuronal circuitries at cellular and sub-cellular levels for extended periods of time[1,2]. Many of these approaches rely on the ability to transport light signals with desired spatial and temporal distributions towards the structures under investigation over large distances inside highly scattering tissue. Multiphoton excitation of fluorescently labelled tissues facilitates good means of imaging of sub-cellular structures to depths of several hundreds of micrometres and resolves individual cells and blood vessels residing in depths in excess of one millimetre[1–6]. At further depths, however, all forms of free-space light-based microscopy suffer from extensive scattering, thereby gradually reducing contrast and resolution. Imaging at these depths is, however, enabled by competing, so-called minimally invasive approaches, essentially a spectrum of endoscopy and endo-microscopy techniques. Nowadays, the most established ones are built upon the ability of gradient-index (GRIN) lenses to relay an image from one of its facets to the other, enabling a variety of imaging modalities to be routinely performed in immobilised as well as motile animal models[7–11].

[1]Institute of Scientific Instruments of the Czech Academy of Sciences, Královopolská 147, 612 64 Brno, Czech Republic. [2]Leibniz Institute of Photonic Technology, Albert-Einstein-Straße 9, 07745 Jena, Germany. [3]Institute of Biophysics of the Czech Academy of Sciences, Královopolská 135, 612 65 Brno, Czech Republic. [4]Institute of Applied Optics, Friedrich Schiller University Jena, Fröbelstieg 1, 07743 Jena, Germany. [5]These authors contributed equally: Miroslav Stibůrek, Petra Ondráčková. ✉e-mail: huhlirova@isibrno.cz; cizmart@isibrno.cz

Recently, a powerful form of endo-microscopy, utilising holographic control of light transport through multimode optical fibres (MMFs), emerged[12–17]. Here, the hair-thin MMF is the only component entering the tissue under investigation, which promises by far the most atraumatic form of deep-tissue observations achievable at unprecedented depths, while minimising blood-brain barrier damage, activation of microglia, neuronal death, and immune response amongst other undesired factors[18,19]. While spatial resolution in GRIN lens-based methods suffers due to optical aberrations (especially when long and narrow elements are used), in the case of MMF endoscopes no such degradation occurs. Here, the spatial resolution derives purely from the used wavelength and the numerical aperture (NA) of the chosen fibre[20].

Despite laser light signals propagating through MMFs being entirely scrambled by the competition of a myriad of MMF modes, the light transport remains deterministic and can be controlled by means of computer holography. The linear operator describing light transport through a specific segment of a MMF is commonly known as the transmission matrix (TM)[21,22]. With its availability one may form any desired optical field leaving the fibre output, within the constraints given by the fibre dimensions and the NA. In imaging applications, the most frequent choice is a sequence of diffraction-limited foci organised across a desired transversal plane[23] raster-scanning the scene pixel by pixel. In practice, each of the desired fields is synthesised by a single digital hologram (kinoform), displayed by a spatial light modulator off which the laser light reflects prior it is coupled into the MMF.

The proof-of-concept demonstrations emerged in 2018 showing structural imaging of neurones in cultures[24], acute brain slices[25,26] and in vivo[24,25,27]. The indication that spatially resolved functional imaging of intracellular calcium may be possible to implement through the means of holographic endoscopy has been shown in[24].

This work pushes these prospects to their actual technological limits and brings them to the forefront of in vivo neuroscience. Through several technological advancements detailed below, the user is given uninterrupted visual feedback as the probe progresses through the brain tissue in search for desired brain structures, individual neurones and their circuits. When convenient, the user can park the probe at a specific location of arbitrary depth and observe neuronal connectivity at the level of dendritic spines[28], the dynamics of sub-cellular structures, neuronal activity and the flow of blood through individual vessels, all with the use of a single instrument.

## Results
Our optical system employs a digital micro-mirror device (DMD) as the holographic modulator (see Fig. 1a). Although this choice is associated with strong losses of optical power, it can provide modulation of amplitude, phase and polarisation of the light signals and achieve the highest purity of foci as well as refresh rate amongst commercially available options[29,30]. Our custom-designed computer interface utilises the full capacity of the DMD's on-board memory and, with the deployment of GPU computing power, allowed us to acquire and utilise TMs, featuring more than one order of magnitude more output channels than reported previously[24,25,27]. This enabled us to deploy multimode optical fibres of much higher information capacity, resulting in 4-fold increase of the observable area simultaneously with 2-fold increase in spatial resolution. Moreover, the computer interface enables efficient calculation of kinoforms for output foci organised across 3-D space (see "Methods" for details), thereby readily applicable for volumetric imaging.

Importantly, we deployed side-view fibre probes[31] in our in vivo experiments, facilitating observation of tissue volumes off the fibre's axis and thereby much less affected by the endoscope insertion (see Fig. 1c and "Methods" for more details). Further, such 'side-ways' observations enabled stitching the image data, obtained as the probe progresses deeper into the brain tissue, into continuous volumetric

records of unprecedented length (see Fig. 1d–h and "Methods" for details of data processing). The versatility of the system is further demonstrated by imaging blood vessels on their own (see Fig. 1i, j) or together with neurone processes (see Fig. 1k) using dual-channel detection. In order to demonstrate the capacity of observing dynamic changes of sub-cellular structures with high resolution, we recorded the dynamics of a punctuate pattern of ZsGreen[32] possibly accumulated in lysosomes[33] migrating within the volume of a cholinergic neurone in amygdala (see Fig. 1l and Supplementary movie 2).

Finally, in Fig. 2 we show how the instrument can be used for high-speed in vivo functional imaging. In the first example, we implement line scans in order to capture spontaneous calcium activity in excitatory neurones expressing GCaMP6s. We note that under the influence of anaesthesia, such calcium activity events become very sparse in the used animal model (See "Methods" for more details on reproducibility of these results). Figure 2a, b shows multiple uncorrelated activity from different cells. Here, one of the neurones is firing with a high frequency while others have very little uncorrelated activity after background subtraction.

In the second example, we show blood flow measurements in a single vessel[34] ≈2mm deep under the brain surface (Fig. 2c–h). Both the time-course (Fig. 2e, g and h) and the power spectrum (Fig. 2f) show characteristic frequencies for breath rate and heart rate. Several measurements also exhibited distinct peaks for low-frequency oscillations below 0.1 Hz (see Supplementary Fig. 4.), which is well known also from functional magnetic resonance imaging across species. It is evoked by the underlying neuronal signalling[35] and plays an important role in studying the resting-state functional connectomics[36,37].

Our last experimental study aspires to provide an indication of the application's impact on tissue stress and the validity of the measured quantities. Using the conventional straight-view regime (bare-terminated fibre, see Fig. 2i) unavoidably affects the tissue by strong mechanical stress, leading to structural and functional changes. The volume in close proximity to the facet, which is most suitable for imaging (the beam degradation due to tissue scattering is here at its lowest), is, sadly, affected most severely. In order to study this problem, we have rebuilt our system for the straight-view regime and set the imaging plane 40 μm away from the fibre facet. While imaging the whole field of view, we gradually inserted the probe into the brain tissue and parked it ≈0.3 mm below the brain surface when a suitable vessel appeared in focus (depth $d$). Here we have initiated blood flow velocity measurements while repeatedly progressing and retracting the fibre, in three steps of 6 μm for each direction (see Fig. 2j). The measured blood flow velocity clearly correlates with the probe movement and its influence grows in highly non-linear manner. We found no possibility to provide the exact equivalent of this study for the side-view regime (see Fig. 2k) and, although it is reasonable to assume that the shape of the probe shall significantly reduce the impact across the imaged volume, we cannot assess how the probe insertion affected the blood flow with respect to its normal state. Yet, we could initiate the blood flow velocity measurement at the proximity of the sharp edge of the probe and, by further increasing its depth, monitor the same vessel at different positions across the field of view. As shown in Fig. 2l, the observed changes in velocity were minuscule, indicating no strong gradients of tissue stress across the field of view. The motion of the side-view probe with respect to the vessel, caused e.g. by an unintentional drift of the system, thereby does not affect the measurement of the blood flow velocity, as it would be in the case of the straight-view alternative.

## Discussion
In summary, we have demonstrated an enhanced imaging capacity of a MMF-based holographic endoscope that enables reaching the imaging quality, stability and speed to resolve the microscopic cornerstones of in vivo neuroscience research: cells, dendrites and spines, sub-cellular

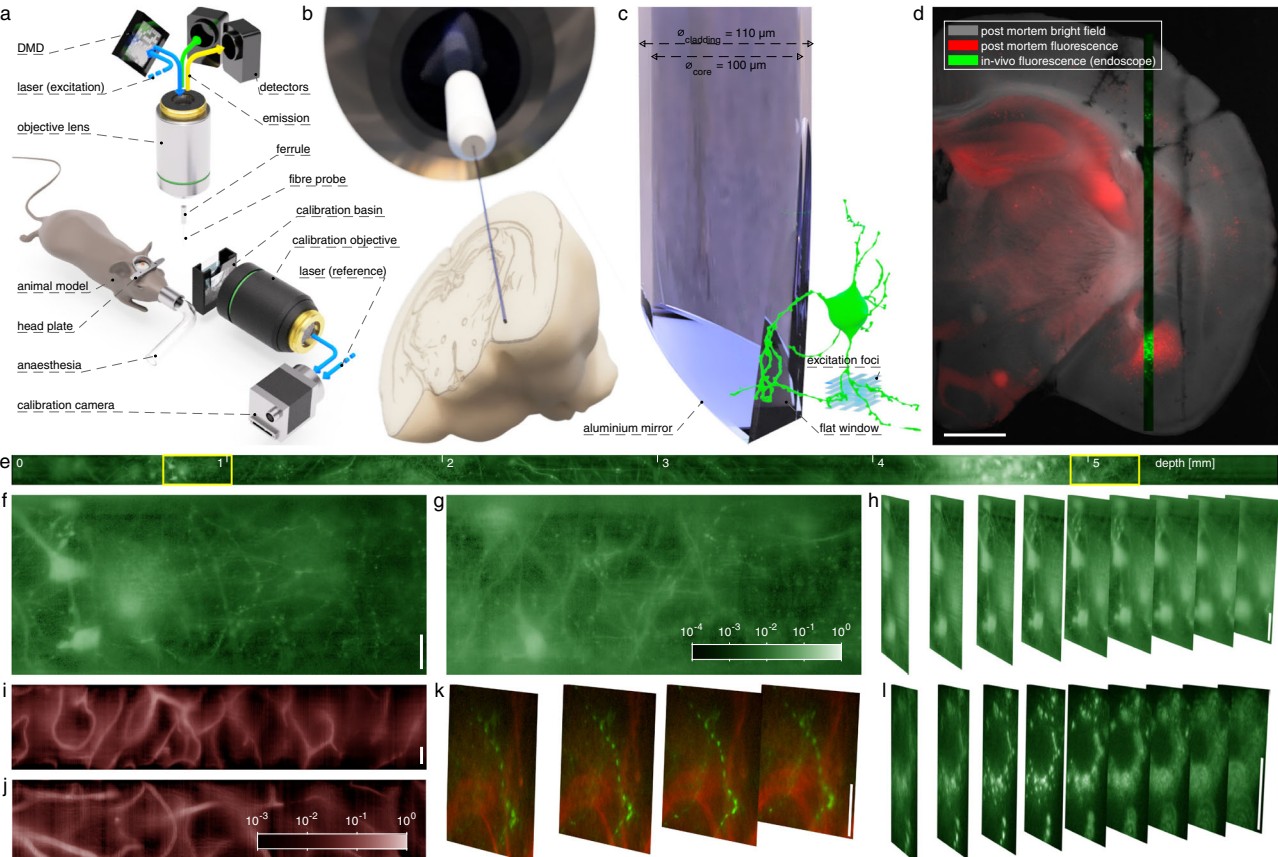

**Fig. 1 | Deep-brain in vivo structure imaging using 110 μm thick MMF endo-microscope. a** Experimental geometry for TM acquisition and imaging (see "Methods" for details). **b** To-scale fibre probe and mouse brain. The brain model has been compiled using Allen Reference Atlas–Adult Mouse[56], available from atlas. brain-map.org. **c** Detail of the side-view fibre probe and 3-D grid of foci scanning the scene. **d** Overlay in vivo MMF endo-microscope record detailed in (**e–h**) with post-mortem bright-field and confocal fluorescence microscopy. Another parallel endoscope insertion trace is visible on the right. **e** Record of endoscope progression (single focal plane set to the distance of 25 μm away from the probe tip) throughout the whole-brain depth of a Thy1-GFP line M mouse (see also Supplementary Movie 1 for volumetric data). **f, g** Details of the same record from the location of cortex and amygdala border, shown in full resolution. **h** Volumetric data corresponding to left part of (**f**) organised in 9 parallel planes 15 μm to 35 μm away

from the probe tip, displaced by 2.5 μm from one another. Fluorescence intensity data in (**e–h**) are shown in logarithmic scale as indicated by a colour bar in (**g**). **i, j** Records of fluorescently labelled (FITC-dextran) blood vessels in cortex area of a wild-type mouse taken 25 μm away from the probe tip (intensity is shown in logarithmic scale as indicated by the colour bar). **k** Dual-channel volumetric imaging of neuronal processes and blood vessel walls (see "Methods" for details of labelling) in Thy1-GFP line M mouse. Focal planes cover an interval of distances from 20 μm to 27.5 μm away from the fibre tip and are displaced 2.5 μm from one another. **l** Volumetric records of sub-cellular structures' dynamics within a single neurone (focal planes organised as in (**h**), see also Supplementary movie 2 and "Methods" for cross-breeding details). The horizontal scale bar in (**d**) corresponds to the length of 1 mm. Vertical scale bars in (**f, h, i, k, l**) correspond to the length of 20 μm.

compartments, spatially resolved intracellular calcium and single-vessel blood flow measurements. All these modalities were implemented through only 110 μm thick optical fibre, free from any additional distal-end optics or scanners, thereby allowing detailed observations anywhere in the brain while greatly preserving functions of important overlying structures (see Supplementary Fig. 5). In addition to the substantial enhancement of the instrument's imaging capacity (spatial resolution and the field of view), we deploy the 'side-view' terminations in in vivo study to provide uninterrupted view throughout the whole depth of the animal brain. Further, image data can be acquired from multiple planes, without the need to provide multiple calibrations, which allows the user to exploit the full potential of volumetric diffraction-limited laser-scanning microscopy.

Many technological and methodological possibilities of this prospect are yet to be exploited; nevertheless, we anticipate that the combination of high imaging quality, unprecedented application depth and uniquely atraumatic nature can greatly complement existing methods of in vivo neuroscience.

Most of the present limitations of the technology are related to the commercially available light modulators. The speed of imaging

(pixel rate), combined with the random access feature is sufficient to enable calcium activity monitoring of potentially hundreds of cells and their processes. Increasing the imaging speed is however desirable if one aspires to further increase the volume of view or to exploit faster fluorescent activity sensors such as those used for voltage imaging[38]. Although we deploy the fastest commercially available light modulation technology (DMD), our pixel rate is at present limited at ≈27 kHz (with active area of the chip reduced to 512 × 512 mirrors). Strategies to enhance the speed performance by orders of magnitude are however already being actively explored[22,39].

The total amount of recorded voxels is currently also limited by the DMD technology: at the maximum speed, the device operates by periodically cycling a sequence of modulations, one for each voxel. The modulations must be pre-loaded and the whole sequence must fit in the on-board memory of the DMD controlling unit. The data-upload rate (USB 3.0) does not currently allow for on-the-fly computer control at the maximum speed. It is however very likely that future generations will overcome this issue, which, next to boosting the imaging capacity, will also enable implementing strategies to allow imaging with flexible fibres discussed below.

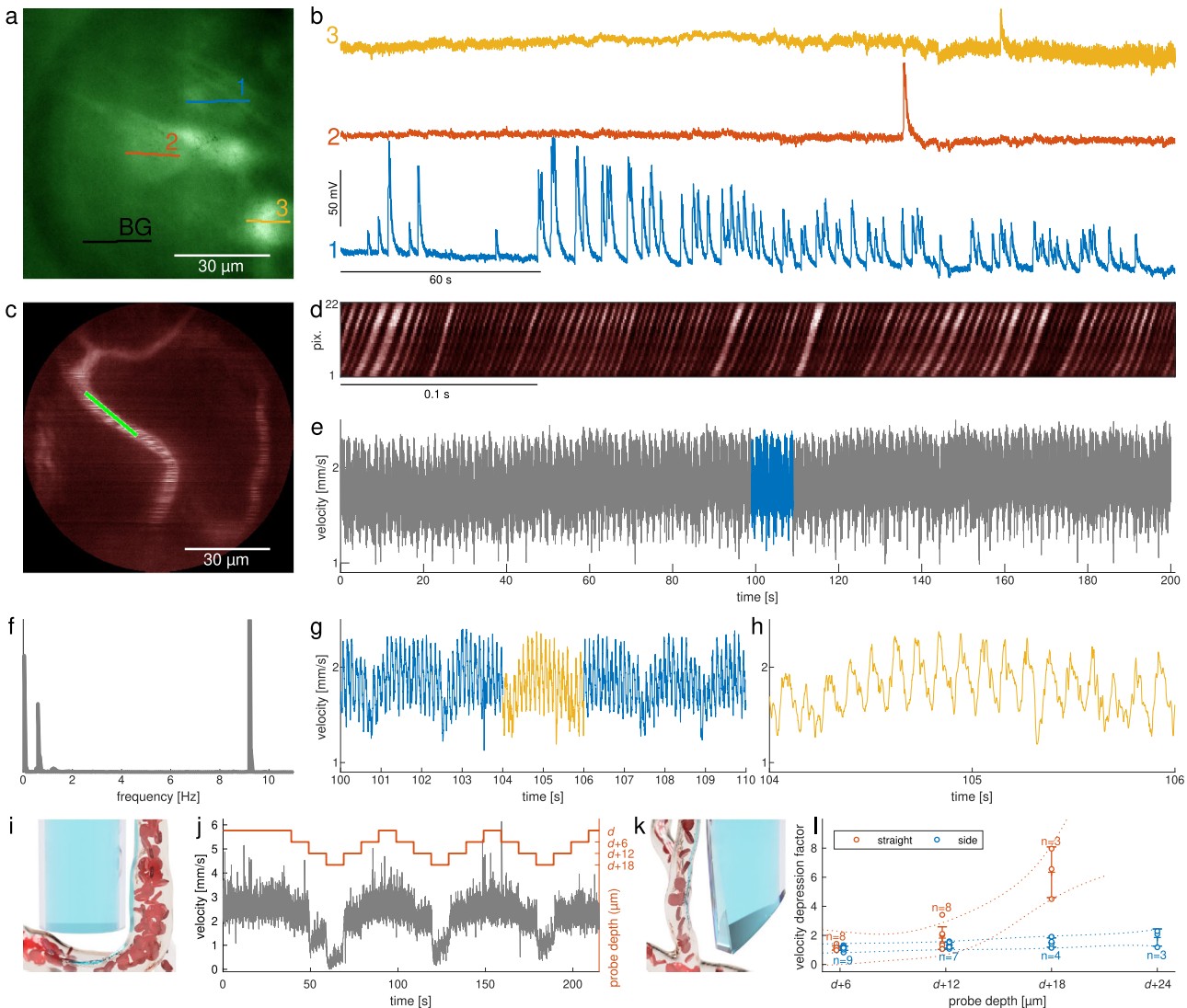

**Fig. 2 | Endoscopic functional imaging. a** A reference image of excitatory neurones expressing GCaMP6s. **b** Spontaneous calcium activity in three regions of interest recorded at ≈60 Hz along straight lines indicated in (**a**) after subtraction of background (recorded along the 'BG' trajectory in (**a**)). **c** A reference image of vasculature stained with FITC-dextran applied i.v. **d** Intensity profiles obtained by scanning along a line indicated in (**c**) at ≈1 kHz showing traces of individual red blood cells. **e**–**h** Reconstructed blood flow velocity, its power spectrum and zoomed-in sections showing characteristic frequencies of breath rate and heart rate. **i** Arrangement of blood flow measurement in the straight-view regime. **j** Blood

flow velocity measured at depth $d$ (fibre facet is ≈40 μm above the imaged vessel), $d + 6$, $d + 12$ and $d + 18$ μm during insertion and retraction of the probe in 3 cycles. **k** Arrangement of blood flow measurement in the side-view regime. **l** Velocity depression (the factor by which the blood flow speed has been reduced due to the vertical movement of the probe) for an interval of probe depths. The error bars indicate standard deviation. The corresponding dotted lines enclose the 99% confidence intervals obtained from the third order polynomial regression of the complete respected datasets, while forcing the model function to pass the point $[d + 0,1]$.

In order to achieve a high purity of foci and thereby high-contrast imagery, the selection of the MMF has to be matched with the imaging capacity of the DMD. Specifically, the number of modes supported by the MMF shall be at least one order of magnitude smaller than the number of mirrors in the active area of the DMD chip. The number of modes grows with the square of the fibre core diameter, which dictates the field of view. At the same time, the number of modes grows with the square of the NA determining both, the spatial resolution as well as the collection efficiency of the fluorescence emission signal. Both quantities are scalable across large intervals: Fibres drawn from glass preforms may reach core diameters in excess of 1 mm. Large field of view is desirable for a number of important reasons, particularly enabling simultaneous observations of remote neuronal circuits and the accumulation of maximum amount of data from single animal. Still, one has to consider that implementation of larger fibres will result in proportionally larger damage to the tissue and thereby affect the

validity of the results as well as the welfare of the animal. The limitation of a small field of view, inherent to minimally invasive fibre-based imaging, has partially been addressed in this work by the possibility to stitch frames obtained along fibre tracks into long records spanning several millimetres. This strategy is particularly suitable for observations of structural connectivity across remote brain regions. Among other strategies to enhance the NA of the optical fibre, exploiting all-solid MMFs made of soft-glass materials has been shown to result in fibres with NA reaching unity[20]. This may lead to achieving almost three-fold improvement in lateral resolution and almost a magnitude enhancement in the axial resolution when compared to these shown in our study. Our selection of MMF, featuring 100 μm fibre core and NA of 0.37 (supporting ≈ 28000 modes), however matches well the bottleneck of the used DMD. We therefore believe this choice highlights the various powers of the current state of the technology optimally. Further, the chosen NA provides more than sufficient collection efficiency

for the system and the used labelling. While the maximum output power at the sample is no larger than 1 mW, due to relatively slow scanning (and thereby long pixel-dwell time) the sensitive detector can easily be saturated when imaging bright somata, therefore we typically reduce the output optical power to 10s–100s of µW.

Owing to appropriate ethical considerations, our technological progress verification in living animals has been conducted acutely under anaesthesia. Only with the successful accomplishments presented here, we regard the technology to be suitably advanced for the first tests in head-fixed, yet awake and behaving animals placed in virtual reality settings, which will facilitate visual cues combined with motion-monitoring treadmill or a floating ball[40], following protocols from similarly invasive experimental approaches[41]. An additional desire in such studies, not achievable with the current application protocols, is returning to the same neuronal circuit in several experimental sessions spanning over large periods of time during which changes due to training or progression of a disease may occur at a cellular and sub-cellular level. We aim to fulfil this aspiration by developing custom connector solutions or exploitations of implantable thin-wall guiding glass tubes[42]. Both will enable the tissue to heal prior to the imaging experiments commence. Moreover, even for the price of a small footprint increase, the later solution will maintain the possibility to stitch neighbouring records and to further broaden the field of view by 360° rotation of the fibre with respect to the animal. While we showcase the approaches in the most available animal models, we see no technological obstacles, which would prevent translating the prospects to large animals, including non-human primates.

Finally, an important and frequently asked question is whether this technology can be translated to allow for observations in freely moving animals. Such behavioural studies would not suffer from biases introduced due to the absence of vestibular cues, which are unavoidable in typical virtual reality experimental settings involving immobilised animals[43,44], and which would be suitable for more complex studies including these of anxiety or social behaviour[45]. Although it has been shown that small bendings (as well as temperature changes) of short fibre probes do not influence the focusing ability significantly[20], high quality imaging over several 100s of mm-long segments of MMF, subjected to deformations expected in experiments involving freely moving animal, remains indeed very challenging. Strategies to provide solutions involve most frequently rapid modifications of the sequence of light modulations according to a TM describing light transport through the actual fibre layout. The actual TM is either selected from numerous TMs obtained upfront experimentally for all expected fibre contortions[46] or calculated on the basis of a theoretical model[16,47,48]. The practical implementation of these ideas relies on the availability of the fast on-the-fly control of the spatial light modulator. Parallel strategies focus on the development of bending resilient MMF, since stark differences of this feature have been identified across different fibre types[49]. It has also been predicted that fibres whose refractive index profile follows the shape of a perfect ellipsoid shall be almost completely immune to bending[50], yet the fibre quality necessary for such applications has not yet been reached. Implementable solutions will most likely involve a combination of both of these prospects.

## Methods
### Experimental system
The complete layout of our experimental setup is available in Supplementary Fig. 1. The whole geometry is similar to the experimental system described in ref. 31. A collimated laser beam with a wavelength of 488 nm (Coherent Sapphire) is divided into a signal and a reference beams. The signal beam is expanded to illuminate the surface of a digital micro-mirror device (DMD) ViALUX V-7001 under an incident angle of ≈24°. The DMD is employed in the off-axis regime, allowing for

phase-only modulation of light[29]. Using two $4f$ systems, the diffracted signal is relayed to the proximal fibre facet and also demagnified to fit the fibre core. The DMD diffracts the desired signal towards two separate zones of the Fourier plane, one for each polarisation state. These are merged using a polarisation beam displacer inside the common path of the relay optics before being coupled into the MMF. The output light reflected from the mirror at the distal end of the side-view probe is imaged on a camera chip (CMOS, Basler) where it interferes with the reference beam. The transmission matrix (TM) is measured using phase-shifting interferometry, which then serves as the basis for calculating the series of holographic modulations (kinoforms) for individual excitation foci.

The TM measurement is essentially analogous to the in situ wavefront correction technique[51], which eliminates all sources of optical aberrations all the way from the source to the detector, thus resulting in the generation of perfect foci. The foci are however optimised at the plane of the camera chip rather than at the desired focal plane. In order to obtain perfect foci at the focal plane of the endoscope (i.e. behind the distal end of the fibre), the system must (i) remain stationary during calibration and consequent imaging and (ii) no aberrations must be present in the optical elements, which are used to image the focal plane onto the calibration camera. Violation of either of these requirements results in optical aberration associated with degradation of the imaging performance. The optical and mechanical components have therefore been chosen and assembled to minimise any thermal and mechanical drift. The whole optical geometry has been placed onto an optical table with active damping of vibrations and the room temperature has been controlled. Further active temperature control has been introduced to the DMD chip[52]. All these factors have resulted in unchanged performance for periods in excess of one day. As the fibre is used in the brain, we have introduced a calibration basin, which is filled with liquid having similar refractive index to that of the brain tissue. The basin's wall forms a microscope cover glass with thickness for which the used calibration objective lens has been optimised (see Fig. 1a). We perform the TM measurement with the fibre focal plane in a close proximity to the inner side of the basin's wall, thereby eliminating the occurrence of a spherical aberration. When imaging in the brain tissue, the only remaining aberrations are introduced in the optical path between the fibre facet and the selected focal plane, by the refractive index inhomogeneities of the tissue itself.

The animal is positioned on a robust, high-precision, 3-axes motorised stage (composed of Physik Instrumente L-511.2ASD00 and Physik Instrumente L-731.44SD), which also houses the imaging components of the calibration module (objective, tube lens, waveplate). This configuration minimises experimental delays and vibrations between calibration and imaging.

The fluorescence signal is collected with the same fibre probe, propagated to the proximal end and spectrally separated from the illumination. If more fluorescent indicators are detected simultaneously, the signal is further separated into two channels.

### Computing interface and calibration procedures
The availability of the transmission matrix (TM) is a requisite aspect of holographic endoscopes. In most practical scenarios it is obtained experimentally by a calibration procedure involving sequential interferometric measurements of conveniently chosen input and output modes[16]. The calibration must be repeated every time a new MMF segment is introduced into the system before imaging can be initiated. For the most common case of laser-scanning endo-microscopy, the last constituent of the calibration procedure requires calculating series of kinoforms based on the acquired TM. When applied on a spatial light modulator located in the optical pathway, each of the kinoforms leads to the formation of a single diffraction-limited focus at the desired position behind the MMF distal facet. The fastest commercially

available holographic modulators are MEMS-based digital micro-mirror devices (DMD) and are able to switch between individual modulations (and corresponding focus positions) at the rate of several 10s of kHz. The calibration procedure of this study follows[27] and uses 21,000 input modes and up to 174,760 output modes, depending on the desired field of view, and limited by the DMD's onboard memory. The TM is measured by a home-built computer interface in National Instruments LabVIEW™, exploiting PC with AMD Ryzen 3700X CPU (8 physical cores) and 128 GB RAM.

In order to achieve volumetric imaging (at various working distances from the distal fibre facet within one scanning sequence), a home-built algorithm exploiting free-space propagation has been developed using low-level code in C programming language and the FFTW library. The shift of a scanning point to a different working distance was achieved using the Fourier-optics-based transformation of the TM. In the first step, the TM describing electromagnetic field in the calibration plane was transformed from the spatial coordinates to the angular spectrum representation[53] using the Fourier transform (FT). Further, the angular spectrum TM was propagated by a refocus distance $z$ (using multiplication by an appropriate spherical phase mask) and transformed back to the spatial coordinates by applying the inverse FT. This transformed TM was consequently used to calculate the kinoforms for the refocused focal planes. To minimise delays, the TM manipulation and kinoforms calculations were performed remotely, in a highly parallelised manner on a high-performance computer, featuring 4 CPUs (Intel Xeon Gold 6126, each 12 physical cores) and 1.5 TB RAM.

## MMF probe design and manufacturing
All probes have been manufactured using a commercially available step-index MMF (CeramOptec Optran Ultra WFGE) having core and cladding diameters of 100 μm and 110 μm, respectively, and 0.37 NA. The lateral spatial resolution (Rayleigh criterion), achievable with this fibre and the wavelength used in this study is 0.8 μm. The fibre has been stripped from the acrylate coating, right-angle cleaved to both extremities at the length of ≈27 mm and glued into a 10.5 mm long ceramic ferrule aligned with the fibre at its proximal end. Signal contamination due to autofluorescence originating from the fibre material is at these length scales negligible[54]. For the straight-view probe, the fibre has not been processed any further. For the side-view probe, the distal fibre facet has been subsequently mechanically polished using an in-house built polishing system under 45° with respect to the fibre axis, rolled around the fibre axis by 180° and polished at 5°, removing part of the cladding and ≈17 μm of the core at the distal facet. The 45° surface has been coated with a reflective aluminium layer using a vacuum evaporator (JEOL JEE-420). Further details are available in[31].

## Animals
All animal experiments were conducted in accordance with protocols approved by the Branch Commission for Animal Welfare of the Ministry of Agriculture of the Czech Republic (permissions no. 47/2020 and 49/2020). All transgenic lines were obtained from Jackson Laboratory, Maine, USA. The Thy1-GFP line M (Stock No: 007788) was used in structural imaging experiments and in dual-colour imaging. Floxed Ai6 (Stock No: 007906), crossed with cre-expressing B6J.ChAT-IRES-Cre (Dneo, Stock No: 031661) were used in imaging of lysosomes.

Floxed Ai162D (Stock No: 031562) crossed with cre-expressing CaMK2a-CreERT2 (Stock No: 012362) treated at the age of 4 weeks with Tamoxifen (75 mg/kg of body weight in a single dose, Sigma-Aldrich) were used for calcium imaging. Wild-type C57BL/6J (Envigo) were used in experiments involving blood flow measurements. Both male and female mice aged 8–24 weeks were used. Mice were housed in a 12 h light /12 h dark cycle at 21 °C to 23 °C and 50–60% relative humidity with ad libitum access to food and water.

## Experimental procedures
**Surgical procedures.** The mice were anaesthetised by inhalation of isoflurane (Vetflurane, Cymedica, Czech Republic, 5% for induction, 1.5% for maintenance) in pure $O_2$ and placed on a portable platform. A warming blanket with a feedback loop from a rectal temperature probe maintained body temperature of 37 °C. After depilation of the fur and local application of lidocaine, skin over the target area was removed by a round incision. The XY coordinates of the target area were identified using the mouse brain atlas[55] and the position of bregma and the target area were marked on the surface of the skull. The 3D-printed custom head plate (see Supplementary Fig. 3) was attached by a cyanoacrylate gel in order to head-fix the animal to the platform. Finally, a high-speed drill (tip diameter 0.3 mm) was used to make a small square craniotomy centred over the target area. For blood flow measurements as well as the studies of blood-brain barrier disruption (see Supplementary Fig. 5), 0.1 ml of 25 μM fluorescein isothiocyanate-dextran (FITC-Dextran mol wt 2,000,000, Merck, USA) was slowly applied to the tail vein of C57BL/6J mice. For two-colour imaging, the blood vessels of Thy1-GFP line M mice were stained by a mixture of PE-labelled anti-endothelium antibodies applied to the tail vein: 50 μl of anti-CD31 (Thermofisher, eBioscience, Cat No. 12-0311-81, Lot 2196690, Clone 390) at dilution of 1:5 and 50 μl of anti-CD34 (Thermofisher, Invitrogen, Cat No. MA5-17831, Lot WE3265594, Clone MEC14.7) at dilution of 1:5. To enhance the fluorescence signal, 100 μl PE-labelled goat anti-rat antibody (Southern Biotech, Cat No. 6420-0, Lot K5813-WI99X) at dilution 1:5 was applied subsequently after 10 min delay and 100 μl PE-labelled donkey anti-goat antibody (Southern Biotech, Cat No 3050-09S, Lot C4208-X888P) at dilution 1:5 was applied with further delay of 10 min. After the end of each experiment, the mice were sacrificed by anaesthesia overdose of ketamine/xylazine (5 mg of ketamine – Bioveta Narkamon 50 mg/ml, and 0.5 mg of xylazine – XYLAZIN Ecuphar 20 mg/ml) administered intraperitoneally and cervical dislocation.

**Image acquisition.** The endoscope is typically calibrated in parallel with the surgical procedures. We set the focal plane commonly 25 μm away from the fibre tip. Additional focal planes are calculated in desired number of steps symmetrically from the calibrated plane, typically in increments of 2.5 μm. When completed, the platform carrying the head-fixed anaesthetised mouse is mounted on the 3D-positioning stage of the endoscope. The stage is used to navigate the probe above the target location within the craniotomy window and to control the insertion of the probe into the brain tissue.

For imaging of tissue along the probe track, we acquired a z-stack for every position of the fibre. A single scan typically takes the form of a horizontally elongated field of view ($100 \times 5.7$ μm$^2$) acquired at different distances from the fibre facet (focal planes) – typically 9 scans, 2.5 μm apart (see supplementary Fig. 2b, c). This way a 3D volume of $100 \times 5.7 \times 22.5$ μm$^3$ is mapped in each z-stack. We scan the z-stacks at 0.3 Hz, while the fibre is penetrating the tissue at a constant speed of 1 μm/s leaving a spatial overlap in each z-stack which allows stitching corresponding frames. This way we obtain an extended 3D volume, which is typically sampled over the extent of $100 \times 6000 \times 22.5$ μm$^3$ i.e. extending throughout the whole-brain depth.

For high-resolution imaging of cells, cellular processes and vessels in a single channel or two channels, we typically scan z-stacks of 4 or 9 focal planes, $57 \times 57$ μm$^2$ or $40 \times 40$ μm$^2$ respectively, 2.5 μm apart (see supplementary Fig. 2a) at a frame rate of ≈ 0.15 Hz.

For calcium as well as blood flow imaging we scan the whole field of view ($100 \times 100$ μm$^2$) in a single focal plane as the probe penetrates the tissue. Once the desired location is reached, the probe motion is stopped, the custom-designed scanning trajectories are defined and time-lapse recording starts. Alternatively, for immediate feedback on Ca$^{2+}$ activity during fibre insertion, scanning along an array of line trajectories covering the whole field of view is used. Once a location exhibiting high Ca$^{2+}$ activity is reached, the probe is parked and a

reference image of the whole FOV is acquired. The time-lapse recording is then started along newly defined trajectories, according to the actual morphology. The scanning frequency is typically in tens of Hz in the case of $Ca^{2+}$ imaging and ≈1 kHz for blood flow.

**Post-mortem imaging.** After completing imaging in the endoscope system, the used transgenic mice were transcardially perfused with 4% paraformaldehyde, the brains were extracted, postfixed for 16 h with 4% paraformaldehyde and 200 μm thick sections were cut with Vibratome (Leica VT1200). Brain slices were then imaged in bright-field and fluorescence confocal modes (using BRUKER Ultima In Vivo Multiphoton system) in order to validate the position of fibre tracks.

### Data processing

**Stitching records along whole-brain depth.** For records of volumes extended along the whole depth of the mouse brain, we use scanning arrangement as shown in Supplementary Fig. 2b, c. In order to enhance the contrast of the whole record, we subtract the mean value of 1% lowest intensity pixels for each scan. Once the probe perforated dura and a small layer of cortex, we assume that constant velocity of the mouse elevation with respect to the probe results in uniform velocity progression of the tissue with respect to the field of view. The tissue gliding around the distal probe tip undergoes a certain amount of compression with a priori unknown gradients, therefore its velocity cannot be reliably derived from the speed of the mouse elevation. We therefore estimate the speed from the mean shift between neighbouring scans (in pixel-per-frame units), utilising the overlapping regions. With the estimate of the speed, we can convert the obtained 4-D record (space and time) in the stitched 3-D record, pixel by pixel. A fraction of pixels is measured more than once (overlapping regions), we therefore use their averaged value in the stitched result.

**Blood flow speed estimation.** Velocity of blood flow was obtained from records of red blood cell traces (see Fig. 2d). The functionality of the used algorithm is outlined in Supplementary Movie 3. The fluorescence signal response visibly differs across rows of the original record (pixels along the scanned trajectory) due to the tissue scattering and noise in the system's calibration. In order to minimise the influence of these issues, the record is corrected demanding the same mean value and standard deviation across each row. To obtain the velocity estimate at a given time, a suitable window of neighbouring line-scans (columns) is used (in the case of data shown in Fig. 2e–h, 4 preceding and 4 following columns were employed). The size of the window has to be adjusted in case the traces are sparse, which leads to low-pass filtering of the results. The window size used in the presented record does not affect frequencies below 100 Hz. The 2-D data array of the selected window is morphed (individual columns are moved downwards or upwards in integer number of pixels) in order to virtually counteract the motion of the cells for the assumed velocity. If the correct velocity is used, the traces must appear horizontally oriented. The averaged standard deviation across the rows of the morphed fields is then used as the metric for the velocity estimation, with the lowest value signifying the best match (2nd order polynomial fit is used for the minimum value and its neighbours).

**Statistics and reproducibility.** The operation of the system is, with the current developmental state, very reliable: technological issues such as fibre breakage, software crush or sudden misalignment due to thermal drift are now extremely rare. Although the number of the in vivo experiments was due to ethical considerations kept at its minimum, it would be safe to state that the system operates with the demonstrated performance in more than 90% of attempts. The only significant factor affecting the successful acquisition of the desired data is the animal model itself. While imaging of structural connectivity and monitoring of blood flow velocity can safely be acquired from each used animal

model, this was not the case for measuring the GCaMP activity where, under the influence of anaesthesia, there was only a very small number of active cells in already limited volume of observation.

We have performed four trials with the fully developed technology, each of which led to the observation of background (neuropile) activity. The clearly spatially resolved activity of at least one neurone in the FOV has been observed in three cases. Only in one attempt, shown in Fig. 2a, b, multiple uncorrelated activity from different cells within the same FOV has been observed.

### Reporting summary

Further information on research design is available in the Nature Portfolio Reporting Summary linked to this article.

## Data availability

The raw datasets generated and analysed during the studies presented in the manuscript have been deposited to the Zenodo repository, https://doi.org/10.5281/zenodo.6598512. The raw dataset which has been used in the Supplementary Information, Supplementary Fig. S4 is available from https://doi.org/10.5281/zenodo.7548492. Mouse Brain Atlas http://labs.gaidi.ca/mouse-brain-atlas has been used to navigate the imaging instrument into the desired location. Figure 1b has been compiled using Allen Reference Atlas−Adult Mouse−Coronal Sections −Average Template, available from atlas.brain-map.org.

## Code availability

Custom scripts written in MATLAB (version R2017b, tested also with R2021a) were used to process all data: The code for stitching of images acquired along the fibre track is available from https://doi.org/10.5281/zenodo.7524794. The code for processing of blood flow velocity is available from https://doi.org/10.5281/zenodo.7524782.

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

## Acknowledgements

The authors would like to acknowledge support from the European Research Council (724530), the Ministry of Education, Youth and Sports of the Czech Republic (CZ.02.1.01/0.0/0.0/15_003/0000476 and CZ.02.1.01/0.0/0.0/16_013/0001775), the European Regional Development Fund (LM2018129), the European Union's H2020-RIA (101016787), Freistaat Thüringen (2018-FGI-0022, 2020-FGI-0032), Institute of Scientific Instruments of the CAS. Thüringer Ministerium für Wirtschaft, Wissenschaft und Digitale Gesell-schaft Thüringer Aufbaubank and Bundesministerium für Bildung und Forschung. We acknowledge the Czech-BioImaging facility ISILMR of the Institute of Scientific Instruments CAS, Brno, Czechia, for hosting our animal experiments. We would like to express our gratitude to Zenon Starčuk jr. and his team for sharing his facilities and assistance with animal models. Further, we would like to thank Ondřej Novák, Hongbo Jia, Sanja Bauer Mikulovic, Janelle Pakan, Dirk Boonzajer-Flaes and Ivo Leite for useful discussions and advice.

## Author contributions

T.Č. conceived the background for the presented technology. M.St., S.T., T.P., H.U. and T.Č. build the experimental geometry. M.St., S.T., M.Ši., T.P., P.J. and A.G. compiled the computer controlling interface. P.O., J.K. and P.K. developed and maintained the used animal lines. T.P., M.St., P.O., P.J. and P.K. developed and manufactured the custom fibre probes. P.O., T.T., P.K and H.U. performed surgeries and post-mortem analysis of the harvested tissues. M.St., P.O., T.T. and T.P. performed all imaging experiments. T.Č. and M.Ši. wrote the scripts for data processing and analysis. H.U. and T.Č. analysed the data, secured funding and led the project. T.Č. wrote the manuscript with contributions from all authors.

## Competing interests

The side-view termination has been patented (DE102021102091) with the following international application being currently examined. S.T., and T.Č. participate in a startup endeavour DeepEn (http://www.deepen.tech), not incorporated at the time of the publication of this article.
