## [Peer Review File · Nature Communications]

110 μ m thin endo-microscope for deep-brain in-vivo observations of neuronal connectivity, activity and blood flow dynamicsREVIEWER COMMENTS

Reviewer #1 (Remarks to the Author):

In this work, the authors present a new imaging technology that enables in-vivo volumetric deep brain imaging. The field of neuroscience would find value in the proposed set up as it offers a resolution below 1µm throughout the whole depth of the mouse brain (which is greater than most current technology) while keeping the diameter of the endoscope to a minimal size and thus preserving brain integrity. It is a very important feature that could facilitate the study of not only deep brain structures in the mouse brain but also in larger animal models.

The manuscript is clearly written, and the authors demonstrate the capabilities of their system by performing well designed experiments. They accomplish experiments that requires high resolution and speed of acquisition (calcium imaging, blood flow velocity), but also other functions that are often not permitted by current technologies such as multi-wavelengths detection, sub-cellular structures imaging. Finally, they test in vivo the design of their side-view probe which they had designed and published earlier and show that it indeed reduces the strength onto the brain tissue. While the side-view mode has also been developed and proven useful with GRIN lenses, the MMF fibre the authors use has the advantage of being significantly smaller.

Questions/remarks to the authors:

1) While this technology seems like it could bring great improvements to current methodologies, it is difficult to judge from the manuscript which of the imaging results are entirely innovative, which have reached their final goal/limitation, which are proofs of concept waiting to be improved. This needs to be clarified and discussed in the manuscript so the readers can understand better the potentials and limitations of the technology.

2) These experiments are performed in (terminally) anesthetized head-fixed mice right after implantation of the multimode optical fibre (MMF). While the need to perform extensive calibration before a new MMF can be used is clearly explained, imaging in these conditions do not seems suitable to many studies. Not only lot of work and animals would be necessary for a single study, but the anaesthetics are likely to affect the physiological functions the researcher would aim to visualise (and particularly the calcium activity of many cell types and the blood flow). The insertion of the fibre is also likely to create some damage and disruption in the surrounding brain tissue. Is it thinkable to do the fibre calibration and then chronically implant the MMF to perform the imaging once the animals have recovered? Could the MMF be placed on a microdrive so it could be moved down on the day of the experiment? Would it/will it be possible to perform these experiments in behaving animals? Or does the bending of the fibre prevent the obtention of the same type of imaging quality?

3) More details about the MMF must be added to the manuscript. While the side-view fibre has been recently published in Siverira et al. (ref 20), this manuscript describes a technique and therefore it is

necessary for the reader to obtain the information right away.

4) The numbers of animals used is not mentioned as it is rightfully explained that the aim of this study was to collect data as proof of concept but it would be useful to know the success rate of such experiments and the difficulties that can be encountered so it can be compared to more traditional methods such as GRIN lens imaging.

Reviewer #2 (Remarks to the Author):

The manuscript on development of a hair-thin laser-scanning endo-microscope enabling in-vivo volumetric imaging for neuron is reviewed.

comments-

1. the authors mentioned that the instrument can perform imaging to whole depth of the mouse brain. however, detail data is not shown.
2. there are several groups and publications towards the proposed study. comment on them publications

Yildirim M, Sugihara H, So PT, Sur M (2019) Functional imaging of visual cortical layers and subplate in awake mice with optimized three-photon microscopy. *Nat Commun* 10(1):177.

Mittmann W, Wallace DJ, Czubayko U, Herb JT, Schaefer AT, Looger LL, Denk W, Kerr JN (2011) Two-photon calcium imaging of evoked activity from L5 somatosensory neurons in vivo. *Nat Neurosci* 14(8):1089–1093.

Kawakami R, Sawada K, Kusama Y, Fang YC, Kanazawa S, Kozawa Y, Sato S, Yokoyama H, Nemoto T (2015) In vivo two-photon imaging of mouse hippocampal neurons in dentate gyrus using a light source based on a high-peak power gain-switched laser diode. *Biomed Opt Express* 6(3):891–901

research groups

<https://scholar.google.com/citations?user=GOWcJagAAAAJ&hl=en>

<https://www.jilab.net/>

https://scholar.google.co.in/citations?hl=en&user=iT1mbHYAAAAJ&view_op=list_works&sortby=pubdate

4. did you observe any autofluorescence signal ?
5. did you consider the optical and sample aberration ?

Reviewer #3 (Remarks to the Author):

NCOMMS-22-18365-T

Tomáš Čížmár

Hair-thin endo-microscope for deep-brain in-vivo observations of neuronal connectivity, activity and blood flow dynamics

The paper describes how a “hair-thin endo-microscope” can be used to explore the innards of a fluorescing mouse brain in vivo and at a moderately high speed. The authors also refer to the instrument as MMF, but I am not sure if they want the community to use this name.

My main comment is that the paper is very well written, the general idea as well as the few applications are outlined and presented in sufficient detail to understand and appreciate the intellectual, academic, and technical efforts required to perform these kind observations. The manuscript relies heavily on the impressive work that has been performed by Tomáš Čížmár, the corresponding author, during the past decade. I am in principle in favor of suggesting the manuscript for publication.

One concern is addressed by the authors, but not satisfactorily. Is the field of view large enough for, e.g., a reasonable contribution to brain research? Is the field of view large enough? What is the cost of increasing it? What would need to be sacrificed? Is this really the kind of microscope that researchers in brain research are waiting for? The authors should discuss these issues in more detail and make sure that they provide a reasonable outlook in the manuscript.

Another concern is, how much damage is afflicted during this kind of observation, i.e., while the fiber is pushed through as well as retracted from the tissue? The fiber may be thin, but it is not so thin that it will not cause some kind of destruction. The authors should provide a very good evaluation of the expected damage, e.g., in terms of loss of previously functional tissue (volume, number of cells, ...). I cannot imagine that they did not already evaluate histological sections.

Furthermore, it is important to provide further details on “side-view fiber probes” and “‘sideways’ observations.” E.g., where is the volume-of-view relative to the front of the fiber? In this context, a question is also, whether it makes sense to, e.g., rotate the fiber and hence explore a larger area or a larger volume in a specimen?

Another important question, which is only partially addressed, is to how much power/energy is the specimen actually exposed? I am not referring to the power/energy that enters the fiber or to losses due to optical effects. Rather, how much power actually exits the fiber and enters the specimen? To how much energy are the specimen and the observed volume exposed?

In this context, it is also important to understand, how efficient the illumination and how efficient the detection processes are. How efficient is the usage of fibers compared with a conventional microscope objective-based fluorescence microscope?

Finally, what are the biggest competitors of this method? Photoacoustic microscopy? Photoacoustic fluorescence micro-endoscopy? Optoacoustic imaging?

REPLY TO REVIEWERS' COMMENTS

We greatly appreciate the thorough assessment of our manuscript, the recognition of its importance and mostly positive comments regarding the clarity of the presented studies. While the manuscript has been invited for major revisions, none of the reviewers have asked us to provide more experimental data to support our conclusions. The reviewers however agree that more discussion is needed to clarify the advantages and limitations of our technology in relevance to the presently available methods. We openly state that our technology has not reached its full power yet and a lot of additional work is still needed to eliminate or at least reduce the impact of the present limitations. Yet, even in its present form the technology offers previously unavailable experimental possibilities. The new version of the manuscript therefore pays much greater attention to this. While most of the relevant information is scattered in our previously published work as well as that of our competitors, we agree that a comprehensive and up-to-date assessment of this technology will add great value to the paper. Our discussion therefore highlights where we see the most perspective utility of the prospect within the field of in-vivo neuroscience and approaches each of the limitations and trade-offs heads on, outlining whether it is likely to persist, whether the solution can be achieved with more advanced technology or whether it is already in our reach.

We believe we have addressed all concerns of the reviewers. Since there are numerous dependencies and trade-offs involved, it is not straightforward to point to specific places in the new text when addressing individual comments, without reciting extensive parts repeatedly in this letter. We would therefore recommend the reviewers to oversee the manuscript text, especially the discussion section prior to returning to their individual reviews.

We believe the new version of the paper will better stimulate the relevant community of technologists to direct their research towards the most important limitations, as well as neuroscientists, which may find that observing details of structural connectivity and activity anywhere in the brain is no longer technologically impossible.

Reviewer #1 (Remarks to the Author):

In this work, the authors present a new imaging technology that enables in-vivo volumetric deep brain imaging. The field of neuroscience would find value in the proposed set up as it offers a resolution below 1 μ m throughout the whole depth of the mouse brain (which is greater than most current technology) while keeping the diameter of the endoscope to a minimal size and thus preserving brain integrity. It is a very important feature that could facilitate the study of not only deep brain structures in the mouse brain but also in larger animal models.

The manuscript is clearly written, and the authors demonstrate the capabilities of their system by performing well designed experiments. They accomplish experiments that requires high resolution and speed of acquisition (calcium imaging, blood flow velocity), but also other functions that are often not permitted by current technologies such as multi-wavelengths detection, sub-cellular structures imaging. Finally, they test in vivo the design of their side-view probe which they had designed and published earlier and show that it indeed reduces the strength onto the brain tissue. While the side-view mode has also been developed and proven useful with GRIN lenses, the MMF fibre the authors use has the advantage of being significantly smaller.

We thank the reviewer for appreciating the importance of our work and the quality of the manuscript's presentation.

Questions/remarks to the authors:

1) While this technology seems like it could bring great improvements to current methodologies, it is difficult to judge from the manuscript which of the imaging results are entirely innovative, which have reached their final goal/limitation, which are proofs of concept waiting to be improved. This needs to be clarified and discussed in the manuscript so the readers can understand better the potentials and limitations of the technology.

The most important outcomes of this work, the state of the technology and the outlook into the future is now clarified in the substantially extended discussion (too large to recite here). It focuses on the limitations of the light-shaping technology and the optical fibre, discussing present and potentially chronic trade-offs.

2) These experiments are performed in (terminally) anesthetized head-fixed mice right after implantation of the multimode optical fibre (MMF). While the need to perform extensive calibration before a new MMF can be used is clearly explained, imaging in these conditions do not seem suitable to many studies. Not only lot of work and animals would be necessary for a single study, but the anaesthetics are likely to affect the physiological functions the researcher would aim to visualise (and particularly the calcium activity of many cell types and the blood flow). The insertion of the fibre is also likely to create some damage and disruption in the surrounding brain tissue. Is it thinkable to do the fibre calibration and then chronically implant the MMF to perform the imaging once the animals have recovered? Could the MMF be placed on a microdrive so it could be moved down on the day of the experiment? Would it/will it be possible to perform these experiments in behaving animals? Or does the bending of the fibre prevent the obtention of the same type of imaging quality?

The calibration does not take a significant amount of time (only about 5-15 minutes, depending on whether single or multiple axial planes are used), which does not limit the time for typical imaging sessions significantly (between 30 minutes and 3 hours). The technology is however ready to be used in awake yet head-fixed animals, provided suitable ethical approvals are available. The protocols for similarly invasive experiments have been published, see e.g. reference [41] of the revised manuscript. Indeed, eliminating the need for acute fibre insertion is highly desirable and we disclose our efforts to provide such solutions via chronically implantable connectors or guiding tubes. Performing such experiments in head-fixed behaving animals is indeed possible and realistic, minuscule bending due to any residual motions won't play an important role. Conducting experiments in freely moving animals would however necessitate much more profound bending which would indeed be challenging to handle. We, together with several research laboratories worldwide devote a lot of effort to solving this problem for a number of years.

All these considerations are now in details explained in the extended discussion.

3) More details about the MMF must be added to the manuscript. While the side-view fibre has been recently published in Siverira et al. (ref 20), this manuscript describes a technique and therefore it is necessary for the reader to obtain the information right away.

The manuscript provides details about the fibre manufacturer and the type, dimensions of the cladding and its core, the resulting NA as well as the dimensions of the termination shape together with a detailed drawing in figure 1c. The complete protocol for the probe manufacturing is detailed in the Methods section, thereby disclosing all information necessary for the reproduction of our results. We have added a link to the Methods section at the location where the probe is mentioned for the first time.

4) *The numbers of animals used is not mentioned as it is rightfully explained that the aim of this study was to collect data as proof of concept but it would be useful to know the success rate of such experiments and the difficulties that can be encountered so it can be compared to more traditional methods such as GRIN lens imaging.*

The operation of the system is, with the current developmental state, very reliable: technological issues such as fibre breakage, software crash or sudden misalignment due to thermal drift are now extremely rare. Although the number of the *in-vivo* experiments was due to ethical considerations kept at its minimum, it would be safe to state that the system operates with the demonstrated performance in more than 90% of attempts. The only significant factor affecting the successful acquisition of the desired data is the animal model itself. While imaging of structural connectivity and monitoring of blood flow velocity can safely be acquired from each used animal model, this was not the case for measuring the GCaMP activity where, under the influence of anaesthesia, there was only a very small number of active cells in already limited volume of observation. With the fully developed technology, this has been attempted using four animals. In all cases we have been able to observe activity of the neuropile, in three cases we have identified activity of at least one cell in the field of view and in one (presented in the manuscript) we have observed the uncorrelated activity of three different cells.

This has been reflected in the manuscript as follows:

“Finally, in Fig. 2 we show how the instrument can be used for high-speed in-vivo functional imaging. In the first example, we implement line scans in order to capture spontaneous calcium activity in excitatory neurones expressing GCaMP6s. We note that under the influence of anaesthesia, such calcium activity events become very sparse in the used animal model. We have performed four trials with the fully developed technology, each of which led to the observation of background (neuropile) activity, yet activity of individual, clearly resolved neurones has been observed in three cases. Only in one attempt, shown in Fig. 2 a–b, a multiple uncorrelated activity from different cells has been observed. Here, one of the neurones is firing with a high frequency while others have very little uncorrelated activity after background subtraction.”

Reviewer #2 (Remarks to the Author):

The manuscript on development of a hair-thin laser-scanning endo-microscope enabling in-vivo volumetric imaging for neuron is reviewed.

comments-

1. the authors mentioned that the instrument can perform imaging to whole depth of the mouse brain. however, detail data is not shown.

We are afraid that the reviewer might have not realised that Fig. 1e corresponds to an *in-vivo* record progression over the whole depth of the mouse brain. The same experimental data is also overlaid in Fig. 1d with the post-mortem bright-field and fluorescence images. There is also the supplementary movie 1 (SM1) associated to Fig. 1e, showing the detailed volumetric record throughout the whole brain depth of a living Thy1-GFP mouse. Further detailed information on the records is presented in the “Image acquisition” subsection in Methods (for example: “an extended 3D volume, which is typically sampled over an extent of $100 \times 6000 \times 22.5 \mu\text{m}^3$ i.e. extending throughout the whole brain depth”), as well as in the “Stitching records along whole-brain depth” subsection in Methods.

Nevertheless, to bring more clarity to the manuscript we have added to the caption of figure 1, a reference to the supplementary movie 1 (SM1) which shows the detailed data on the volumetric record over the whole depth of mouse brain. The following sentence was added:

“Fig. 1. (...) e, Record of endoscope progression (single focal plane set to a distance of 25 μm away from the probe tip) throughout the whole brain depth of a Thy1-GFP line M mouse (see also Supplementary movie SM1 for volumetric data).”.

2. there are several groups and publications towards the proposed study. comment on them publications

Although all the proposed publications and groups deal with *in-vivo* imaging of neurones, none of them refers to the presented method, i.e. exploitation of holographically controlled light delivery through multimode fibre optics and particularly its unique feature of accessing the deepest regions of the brain at sub-micron resolution performance. Our manuscript states early that it aspires to offer solutions beyond the capabilities of the mentioned methods (non-invasive multiphoton microscopy and optimised GRIN-lens based endoscopy). Therefore, we would like to progress to the actual topic (submicron imaging deeper than multiphoton microscopy, with footprint and resolution better than any GRIN lens-based approaches) without extensive dwelling on the established methods in excessive details as they are notoriously well known in the discipline. All the suggested references are included in the relevant part of the introduction and beyond, which now specifically highlights multi-photon microscopy amongst the free space techniques.

Yildirim M, Sugihara H, So PT, Sur M (2019) Functional imaging of visual cortical layers and subplate in awake mice with optimized three-photon microscopy. Nat Commun 10(1):177.

The work is now included as reference [3].

Mittmann W, Wallace DJ, Czubayko U, Herb JT, Schaefer AT, Looger LL, Denk W, Kerr JN (2011) Two-photon calcium imaging of evoked activity from L5 somatosensory neurons in vivo. Nat Neurosci 14(8):1089–1093.

The work is now included as reference [4].

Kawakami R, Sawada K, Kusama Y, Fang YC, Kanazawa S, Kozawa Y, Sato S, Yokoyama H, Nemoto T (2015) In vivo two-photon imaging of mouse hippocampal neurons in dentate gyrus using a light source based on a high-peak power gain-switched laser diode. Biomed Opt Express 6(3):891–901

The work is now included as reference [5].

research groups

<https://scholar.google.com/citations?user=G0WcJaqAAAAJ&hl=en>

One work of Rafael Yuste is cited as reference [1] as an example of the discipline’s highlights. Further his earlier work is cited when introducing dendritic spines as reference [28].

<https://www.jilab.net/>

Na Ji contributes to the development of non-invasive multiphoton approaches as well as the minimally invasive GRIN-lens-based methods. Both efforts are now mentioned as references [6] and [7] respectively.

https://scholar.google.co.in/citations?hl=en&user=iT1mbHYAAAAJ&view_op=list_works&sortby=pub

date

Similarly, Tommaso Fellin exploits non-invasive as well as minimally invasive. His work is cited as reference [2] and [11].

4. did you observe any autofluorescence signal ?

Autofluorescence in our geometry is far below an observable level. The signal is expected to be less than 10 orders of magnitude lower than the excitation power (see e.g. [M. Bianco et al. Biomed. Opt. Express 12, 993-1009 (2021)], presenting a rigorous measurement of autofluorescence level for photometry probes of identical chemical content and similar geometry – now cited as [54]).

This is now mentioned in the Methods section:

“... The fibre has been stripped from the acrylate coating, right-angle cleaved to both extremities at the length of ~27 mm and glued into a 10.5 mm-long ceramic ferrule aligned with the fibre at its proximal end. Signal contamination due to autofluorescence originating from the fibre material is at these length scales negligible[54]. For the straight-view...”

5. did you consider the optical and sample aberration ?

As for the optical aberrations, we have attempted to eliminate all thinkable sources of their origin. This is now detailed in the Methods section as follows:

“... The transmission matrix (TM) is measured using phase-shifting interferometry, which then serves as the basis for calculating the series of holographic modulations (kinoforms) for individual excitation foci.

The TM measurement is essentially analogous to the in-situ wavefront correction technique[51], which eliminates all sources of optical aberrations all the way from the source to the detector, thus resulting in the generation of perfect foci. The foci are however optimised at the plane of the camera chip rather than the desired focal plane. In order to obtain perfect foci at the focal plane of the endoscope (i.e. behind the distal end of the fibre), the system must (i) remain stationary during calibration and consequent imaging and (ii) no aberrations must be present in the optical elements, which are used to image the focal plane onto the calibration camera. Violation of either of these requirements results in optical aberration associated with degradation of the imaging performance. The optical and mechanical components have therefore been chosen and assembled to minimise any thermal and mechanical drift. The whole optical geometry has been placed onto an optical table with active dumping of vibrations and the room temperature has been controlled. Further active temperature control has been introduced to the DMD chip[52]. All these factors have resulted in unchanged performance for periods in excess of one day. As the fibre is used in the brain, we have introduced a calibration basin, which is filled with liquid having similar refractive index to that of the brain tissue. The basin's wall forms a microscope cover glass with thickness for which the used calibration objective lens has been optimised (see Fig. 1a). We perform the TM measurement with the fibre focal plane in a close proximity to the inner side of the basin's wall, thereby eliminating the occurrence of a spherical aberration. “

As for the sample aberrations, they can only origin from the tissue in-between the fibre facet and the selected focal plane, usually between 15 and 35 μm . Here, the light transport is associated with the same scattering phenomena as in any free-space microscopy technique. The quality of our results shows that at these distances the degradation does not prevent one to observe the sub-cellular features. We continue the method section as follows:

“When imaging in the brain tissue, the only remaining aberrations are introduced in the optical path between the fibre facet and the selected focal plane, by the refractive index inhomogeneities of the tissue itself.

The animal is positioned on a robust, ...”

Reviewer #3 (Remarks to the Author):

NCOMMS-22-18365-T

Tomáš Čížmár

Hair-thin endo-microscope for deep-brain in-vivo observations of neuronal connectivity, activity and blood flow dynamics

The paper describes how a “hair-thin endo-microscope” can be used to explore the innards of a fluorescing mouse brain in vivo and at a moderately high speed. The authors also refer to the instrument as MMF, but I am not sure if they want the community to use this name.

My main comment is that the paper is very well written, the general idea as well as the few applications are outlined and presented in sufficient detail to understand and appreciate the intellectual, academic, and technical efforts required to perform these kind observations. The manuscript relies heavily on the impressive work that has been performed by Tomáš Čížmár, the corresponding author, during the past decade. I am in principle in favor of suggesting the manuscript for publication.

We thank the reviewer for the support.

One concern is addressed by the authors, but not satisfactorily. Is the field of view large enough for, e.g., a reasonable contribution to brain research? Is the field of view large enough? What is the cost of increasing it? What would need to be sacrificed? Is this really the kind of microscope that researchers in brain research are waiting for? The authors should discuss these issues in more detail and make sure that they provide a reasonable outlook in the manuscript.

There are several factors limiting the field of view intertwined in a complex interplay of compromises one must consider. This is now explained in detail in the discussion, too robust to recite here. As presented in the manuscript, stitching observations into one large field of view is one possible route to deal with this limitation, but it is not applicable when fast changes are to be observed. Larger field of view can always be achieved using larger fibres which are readily available, yet one must count with more severe invasiveness and, at least with the performance of the present DMD, one must sacrifice either the resolution, the contrast, or the speed.

Another concern is, how much damage is afflicted during this kind of observation, i.e., while the fiber is pushed through as well as retracted from the tissue? The fiber may be thin, but it is not so thin that it will not cause some kind of destruction. The authors should provide a very good evaluation of the expected damage, e.g., in terms of loss of previously functional tissue (volume, number of cells, ...). I cannot imagine that they did not already evaluate histological sections.

As in-vivo optogenetics and photometry uses essentially fibres of the same thickness in awake animal models, there is ample overview regarding these considerations in the available literature. We now cite the following works to address this issue:

18. Gulino, M., Kim, D., Pané, S., Santos, S. D. & Pego, A. P. Tissue response to neural implants: The use of model systems toward new design solutions of implantable microelectrodes. *Frontiers in Neuroscience* 13 (2019).
19. Kozaj, T. D. Y., Jaquins-Gerstl, A. S., Vazquez, A. L., Michael, A. C. & Cui, X. T. Brain tissue responses to neural implants impact signal sensitivity and intervention strategies. *ACS Chemical Neuroscience* 6, 48–67 (2015).

Further, we indeed perform post-mortem analysis in all our experiments routinely. The most standard (H&E) histology does not seem to provide the best evidence of the tissue damage, see an example below:

Therefore, we offer a new supplementary material in Fig. S5, essentially a study of both insertion and retraction of the probe, outlining that most bleeding occurs after the acute imaging.

Furthermore, it is important to provide further details on “side-view fiber probes” and “sideways’ observations.” E.g., where is the volume-of-view relative to the front of the fiber? In this context, a question is also, whether it makes sense to, e.g., rotate the fiber and hence explore a larger area or a larger volume in a specimen?

The volume dimensions and its position with respect to the fibre is now clearly stated in the caption of the figure 1 as well as in the Methods section. The possibility to exploit 360° rotation (in combination with thin-wall guiding tube) is included in the discussion.

Another important question, which is only partially addressed, is to how much power/energy is the specimen actually exposed? I am not referring to the power/energy that enters the fiber or to losses

due to optical effects. Rather, how much power actually exits the fiber and enters the specimen? To how much energy are the specimen and the observed volume exposed?

This is now also covered in the discussion. Due to the relatively slow pixel rate (the DMD refresh rate), the pixel dwell time is much larger when compared to typical laser-scanning microscopy utilising galvo or resonator scanners. Therefore, even with lower collection efficiency we can keep the power exposure below one milliwatt while maintaining very good signal to noise ratio.

In this context, it is also important to understand, how efficient the illumination and how efficient the detection processes are. How efficient is the usage of fibers compared with a conventional microscope objective-based fluorescence microscope?

This is now also included in the discussion. Briefly, the collection efficiency scales with the square of the NA. The fibre probe's NA of 0.37 has therefore roughly between 15% and 40% of the collection efficiency when compared with laser scanning microscopes, but due to the abundant pixel dwell time, our demonstrations do not suffer from a shortage of available signals.

Finally, what are the biggest competitors of this method? Photoacoustic microscopy? Photoacoustic fluorescence micro-endoscopy? Optoacoustic imaging?

It would be hard to draw a line across the whole spectrum of imaging approaches used in brain observations. As we mainly focus on fluorescently labelled cells and large depths inside the tissue, we would consider the closest competitors to be free-space multiphoton microscopy and GRIN-lens based laser-scanning or wide-field endo-microscopy approaches, both now better introduced as the current state-of-the-art in the manuscript. The photo/opto-acoustic microscopy might add additional modality (as seen for example in recent works by the team of E. Bossy), yet there are numerous other modalities such as CARS, SHG, super-resolution techniques based on compressive sensing or non-linear response, which would deserve equal attention. Therefore, we would prefer not to branch the discussion beyond pure fluorescence-based techniques.

REVIEWERS' COMMENTS

Reviewer #1 (Remarks to the Author):

The authors have addressed all of my comments. The new and extended discussion is a particularly welcome addition that highlights the technological limitations of the technique but also possible solutions. It will strengthen the relevance and interest of the manuscript. I recommend this paper for publication.

Reviewer #2 (Remarks to the Author):

The author addressed all the comments raised. It can be accepted in present form.

Reviewer #3 (Remarks to the Author):

I am satisfied with the many improvements made by the authors and suggest the manuscript's publication.

REPLY TO REVIEWERS' COMMENTS

Reviewer #1 (Remarks to the Author):

The authors have addressed all of my comments. The new and extended discussion is a particularly welcome addition that highlights the technological limitations of the technique but also possible solutions. It will strengthen the relevance and interest of the manuscript. I recommend this paper for publication.

Reviewer #2 (Remarks to the Author):

The author addressed all the comments raised. It can be accepted in present form.

Reviewer #3 (Remarks to the Author):

I am satisfied with the many improvements made by the authors and suggest the manuscript's publication.

There is nothing left to say except expressing our gratitude to all reviewers for recognizing the thorough efforts we have invested in improving the article.